# Complex Cd–Pb Multigenerational Exposure Improves the Growth and Food Utilization of the Cutworm *Spodoptera litura*

**Meifang Zhang** [1,2,3,†], **Jin Chen** [1,2,3,4,†], **Kaixuan Wu** [1,2,3], **Shiru Chen** [1,2,3], **Jianwu Wang** [1,2,3] and **Yinghua Shu** [1,2,3,*]

1  Key Laboratory of Agro-Environment in the Tropics, Ministry of Agriculture, South China Agricultural University, Guangzhou 510642, China
2  Guangdong Engineering Research Centre for Modern Eco-Agriculture, Guangzhou 510642, China
3  Department of Ecology, College of Natural Resources and Environment, South China Agricultural University, Guangzhou 510642, China
4  School of Life Sciences, Guizhou Normal University, Guiyang 550025, China
*  Correspondence: shuyinghua@scau.edu.cn
†  These authors contributed equally to this work.

**Abstract:** Concurrent exposure to cadmium (Cd) and lead (Pb) is prevalent in the environment, but information on the long-term impacts of complex Cd–Pb exposure on herbivorous insects, especially at low doses, is scant. We studied the effects of complex Cd–Pb exposure (4.06 mg/kg Cd and 12.5 mg/kg Pb) on the growth and food utilization of the herbivorous insect *Spodoptera litura* for 10 continuous generations. Cd or Pb ingestion, excretion and accumulation by insect at the different developmental stages was determined for ten generations. The weighted scores calculated by the analytic hierarchy process (AHP) on the basis of the parameters of survival, growth and food utilization indicated that complex Cd–Pb exposure had positive impacts on the insects, regardless of generations. Compared with Cd (4.06 mg/kg Cd) and Pb (12.5 mg/kg Pb) alone, complex Cd–Pb exposure showed antagonistic interactions. After exposure to complex Cd–Pb, although larvae significantly increased heavy metal uptake along with the elevated ECD and ECI, insects enhanced heavy metal excretion via the feces and puparium; consequently, heavy metal accumulation in the insect body significantly decreased. With increasing generations, the positive impacts of complex Cd–Pb stress on insects became increasingly obvious, and the uptake of Cd or Pb decreased while the excretion of Cd or Pb increased, which demonstrated that insect tolerance to Cd–Pb exposure at low concentrations increased over generations.

**Keywords:** accumulation; complex heavy metal stress; herbivorous insects; metal tolerance; multigeneration; nutritional indices

**Key Contribution:** Complex Cd–Pb exposure at low concentrations had positive effects on insect growth and food utilization, which enhanced over generations. Compared with Cd and Pb alone, complex Cd–Pb exposure at low concentrations showed antagonistic effects on insects.

## 1. Introduction

Humans and their biota have been chronically exposed to thousands of pollutants, including heavy metals [1–3]. These pollutants are often found as mixtures in the environment [4], potentially resulting in additive, antagonistic or synergistic effects due to differences in bioavailability and affinity [5], differences in the target organisms, exposure duration, concentration, and the nature of the pollutants [6–10].

Currently, heavy metal pollution has become an increasingly important global ecological concern [11–15]. Cadmium (Cd) and lead (Pb) are two common heavy metals in soil [16] that have a high degree of toxicity and have been designated as the priority heavy metals to control in China [17]. Both Cd and Pb have also been shown to cause serious damage

to the survival, development, growth, reproduction, and behavior of organisms [18–22]. Cd and Pb are often present together [23], and contamination by mixtures of Pb and Cd is prevalent in the soil of the Pearl River Delta region [24,25]. Moreover, Wu et al. showed that the combined effects of Pb and Cd were very complex [26].

Herbivorous insects, which serve an important role in ecosystems as a link between the base of the food web (e.g., vegetables and crops) and higher order consumers (e.g., predators, birds, chickens), are the core of the bioaccumulation and transfer of heavy metals in the food chain and food web [27,28]. Thus, the study of the impact of the accumulation in herbivorous insects on their growth and other activities is of significance. The impacts of single heavy metal stress on insects have been well studied [29], and high heavy metal stress evidently inhibited their survival, growth and reproduction [30–35]. Long term (low level) exposure to heavy metals improves insect tolerance to heavy metals [36–39] and even improves their tolerance to other stressors (e.g., pesticides) [40,41]. However, these recent studies were carried out in the short term, ignoring the long-term and cumulative effects of heavy metals. Moreover, the information on the toxicity of heavy metal mixtures on insects, especially at low doses, is scant [5,29].

The common cutworm, *Spodoptera litura* Fabricius (Lepidoptera, Noctuidae), is one of the most common herbivorous insects widely distributed in the world [42]. In natural conditions, *S. litura* larvae feed mainly on crops such as cotton, soybean, groundnut, tobacco, and then pupate in the soil of farmland [42]. Thus, they are mainly exposed to heavy metals through their food-plants and soils [43]. Since a preferable artificial diet was supplied for *S. litura* for several years [43], we carried out a series of insect bioassays and analyses using a heavy metal contaminated artificial diet. In previous studies, we demonstrated that single Cd or Pb stress below the tolerance threshold level (4.06 mg/kg Cd [44]; 12.5 mg/kg Pb [30,45]) had a positive or no effect on growth, development and reproduction of *S. litura*. Therefore, in this study, we tested single Cd stress (4.06 mg/kg Cd), single Pb stress (12.5 mg/kg Pb) and complex Cd–Pb stress (4.06 mg/kg Cd and 12.5 mg/kg Pb) to feed *S. litura* for ten continuous generations under laboratory conditions. We aimed to investigate Cd or Pb ingestion, excretion and accumulation in *S. litura* at the different developmental stages and analyze their combined effects on growth and food utilization after exposure to complex Cd–Pb stress in the long term.

## 2. Materials and Methods

### 2.1. S. litura Rearing and Complex Cd–Pb Treatment

Eggs of common cutworm *Spodoptera litura*, the larvae of which were reared on the standard artificial diet for various generations [29,46], were from the Insectarium of the Department of Ecology in South China Agriculture University. The artificial diet was designed using the ingredients described previously [46]. The detailed ingredients are shown in the Supplementary Materials (Table S1). Each diet was identical, expect for differences in heavy metals.

Upon hatching, the larvae were fed the above artificial diet supplemented with different levels of $CdCl_2$ and $Pb(NO_3)_2$ (Merck, Darmstadt, Germany). The final concentrations of heavy metals added to the diet were 0 (CK), 4.06 mg/kg Cd (denoted as Cd), 12.5 mg/kg Pb (Pb) and 4.06 mg/kg Cd +12.5 mg/kg Pb (M). When pupae became adults, pairs of healthy and vibrant adults from each treatment were transferred to a special adult jar pasted with egg laying paper. In the central of jar bottom, a ball of cotton soaked in honey water was placed for the adults. Eggs laid by adults from the first generation from each treatment were considered as the beginning of the next generation, and other generations were treated in the same way. The rearing conditions were described in the Supplementary Materials (Text S1).

### 2.2. S. litura Bioassay with Complex Cd–Pb Stress for the First, Fifth and Tenth Generations

Larvae were treated as described in Section 2.1 and were reared individually as described in [46]. Fifty individuals were used for one bioassay and each set of bioassays

were repeated in triplicate. The survival rate of *S. litura* larvae (%), the pupation rate (%) and eclosion rate (%) were calculated as described in [46]. The body of *S. litura* at different developmental stages (larvae, pupae, and adults) was weighed and recorded. The determination and calculation formulae were described in the Supplementary Materials (Text S2 and Table S2).

### 2.3. S. litura Nutritional Indices with Complex Cd–Pb Stress for the First, Fifth and Tenth Generations

Larvae were treated and reared individually as described in Section 2.2. Fourteen individuals were used for one bioassay and each set of bioassays was replicated in triplicate. Nutritional indices, an understanding of the processes involved in Lepidoptera larval nutrition [47], including relative consumption rate (RCR) (g/g/d), approximate digestibility (AD) (%), efficiency of conversion of ingested food (ECI) (%), efficiency of conversion of digested food (ECD) (%), relative growth rate (RGR) (g/g/d), were determined and calculated as described [47]. The calculation formulae are shown in the Supplementary Materials (Table S2).

### 2.4. Measurement of Pb and Cd Concentrations

The larvae were treated and reared individually as described in Section 2.2. For the first, fifth and tenth generations, the 6th instar larvae, prepupae, pupae, and adults (24 insects per treatment) were sampled, with four replicates. After larvae were dissected, their bodies for all replicates were sampled. Additionally, the feces and exuviates from the corresponding treatments at the different developmental stages were sampled. The puparium was also sampled. Cd or Pb concentrations were measured as described in our previous study [46]. Cd or Pb ingestion, excretion, and accumulation by *S. litura* at different developmental stages were calculated as described [46]. The detailed determination and calculation methods are described in the Supplementary Materials (Text S3).

### 2.5. Statistical Analysis

Statistical analyses were performed using the software package SPSS (version 22; SPSS, Inc., Chicago, IL, USA). After testing data for normality using descriptive statistics followed by exploration, generalized linear models (GzLMs) and one-way analyses of variance (ANOVAs) followed by Tukey's HSD test were used to analyze significant differences. The Pearson correlation coefficient was used to analyze the correlations between Cd or Pb accumulation in insects and biological parameters and nutritional indices. The percentages of data were arcsine square root transformed before the analysis. Other data were log transformed when necessary to verify variance homogeneity.

To intuitively present the impact of heavy metal stress on the life characteristics and food utilization of *S. litura* during continuous multi-generations, the analytic hierarchy process (AHP) [48] was used. Taking the survival, weight and food utilization of *S. litura* as the decision goals, which were then subdivided into specific elements, the corresponding weight values of the decision goal and element were given by the expert grading method (Supplementary Data Table S3). The corresponding score for each element was given in line with the results of one-way ANOVA (Supplementary Data Table S4), and the weighted total score was calculated.

## 3. Results
### 3.1. Impacts of Complex Cd–Pb Exposure on Survival, Growth and Food Utilization of S. litura
3.1.1. Impacts of Complex Cd–Pb Exposure on Survival and Growth of *S. litura*

The complex Cd–Pb exposure had no effects on the survival of *S. litura* during the different developmental stages, regardless of the generation (Figure 1A–C). Specifically, after exposure to Pb, a significantly reduced eclosion rate was found in insects from the first generation (Figure 1C).

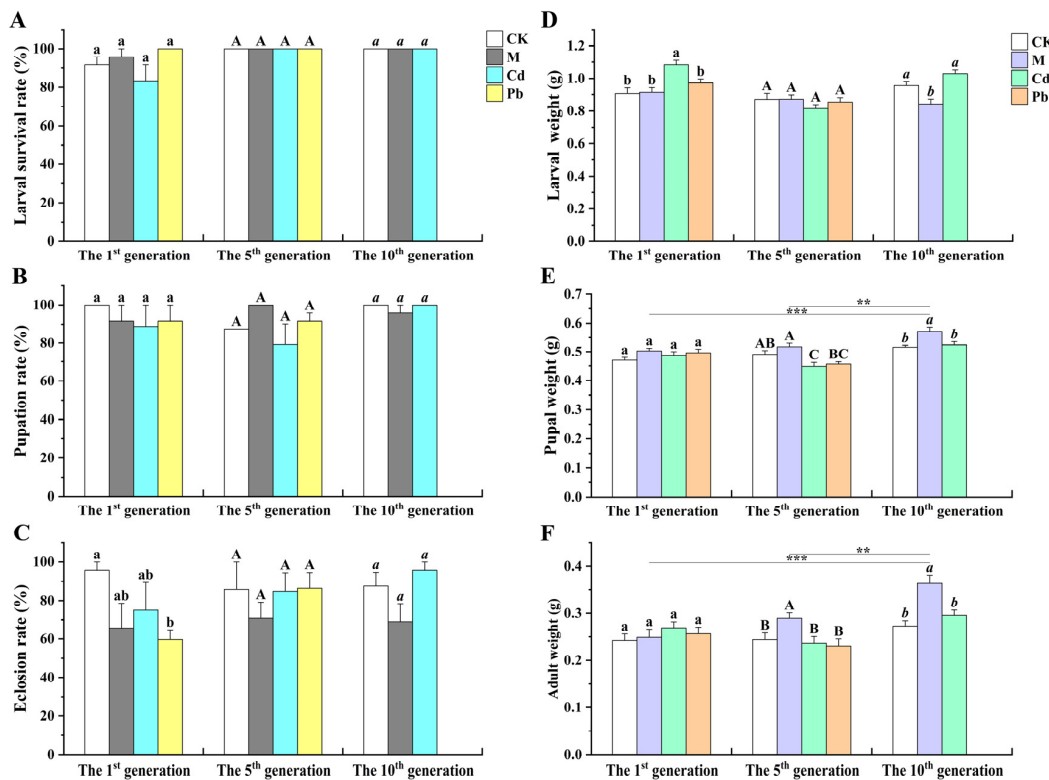

**Figure 1.** Impacts of complex Cd–Pb exposure on biological parameters of *S. litura*. After *S. litura* was exposed to heavy metal (0 (CK), 4.06 mg/kg Cd +12.5 mg/kg Pb (M), 4.06 mg/kg Cd (Cd) and 12.5 mg/kg Pb (Pb)): the survival rate of *S. litura* larvae (**A**), pupation rate (**B**), eclosion rate (**C**); larval weight (**D**), pupal weight (**E**), adult weight (**F**). Lowercase, uppercase and italic letters indicate significant differences among treatments in the same generation, and the asterisk (*) indicates significant differences between generations under complex Cd–Pb exposure (** $p < 0.01$, *** $p < 0.001$).

After Pb exposure for eight generations, significant malformation was observed in pupae, which failed to reach adulthood (Supplementary Data Figure S1). Thus, we did not obtain data on *S. litura* from the tenth generation exposure to Pb. After exposure to complex Cd–Pb for ten generations, the larval weight was significantly lower than that in the control and single Cd treatment groups (Figure 1D). However, a significantly higher larval weight was found in Cd-treated insects from the first generation (Figure 1D). Complex Cd–Pb exposure did not affect the pupal weight of insects from the first and fifth generations, but a significantly increased pupal weight was detected in insects from the tenth generation (Figure 1E). After exposure to complex Cd–Pb for five and ten generations, the weight of adults significantly increased compared with the control and single Cd or Pb treatments, and a significant difference was found between the first, fifth and tenth generations (Figure 1F).

GzLMs results revealed that the generation number had a strongly significant effect on the larval survival rate and weight (Supplementary Data Table S5). Except for the pupation rate, treatment had a strongly significant effect on life history traits. Additionally, the interactions between generation and treatment on the larval survival rate and the weight of larvae, pupae and adults were statistically significant.

### 3.1.2. Impacts of Complex Cd–Pb Exposure on Food Utilization of *S. litura* Larvae

The complex Cd–Pb exposure did not impact the approximate digestibility (AD) and efficiency of conversion of digested food (ECD) of *S. litura*, a significantly decreased AD and increased ECD occurred over generations (Figure 2A,C). A significantly higher efficiency of conversion of ingested food (ECI) was found in complex Cd–Pb-treated *S. litura* from the first and tenth generations, with a significantly increasing trend over generations (Figure 2B). After

exposure to complex Cd–Pb, insect relative consumption rate (RCR) significantly decreased with generations, with the significantly lower RCR from the first and tenth generations when compared with control insect (Figure 2D). Specifically, significantly higher relative growth rate (RGR) was observed in complex Cd–Pb-treated larvae than in control larvae from the first generation, and the opposite was found in complex Cd–Pb-treated larvae from the fifth and tenth generations (Figure 2E). Generation and treatment had a strongly significant effect on the food utilization of *S. litura*, and their interactions also showed a significant effect on RGR, AD and ECD (Supplementary Data Table S5).

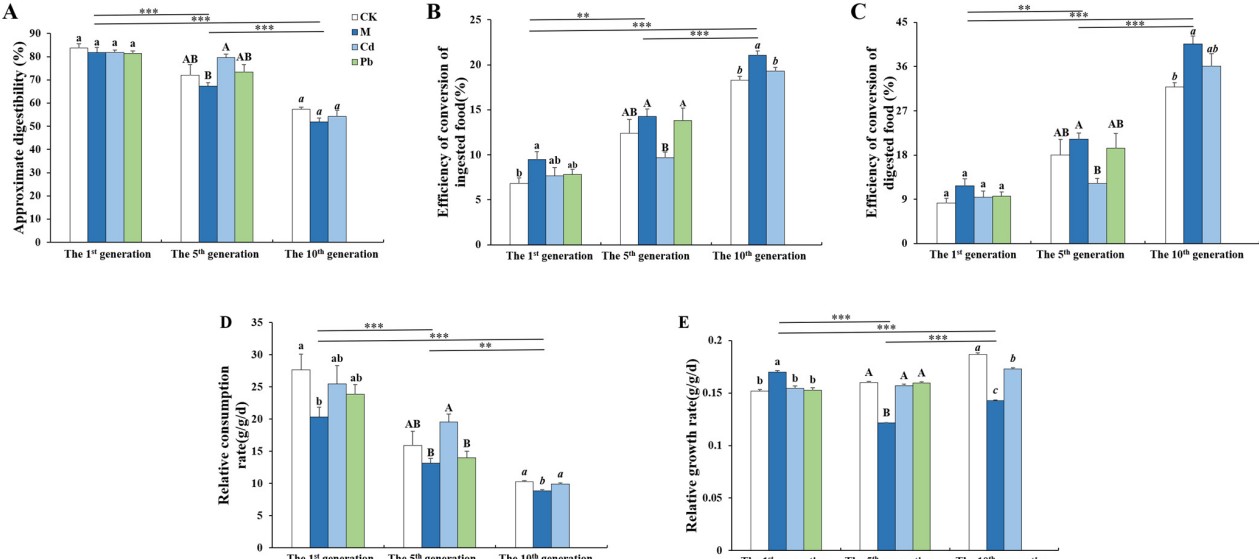

**Figure 2.** Impacts of complex Cd–Pb exposure on food utilization of *S. litura* larvae. Approximate digestibility (**A**), Efficiency of conversion of ingested food (**B**), Efficiency of conversion of digested food (**C**), Relative consumption rate (**D**), Relative growth rate (**E**) of *S. litura*. From left to right are the first, fifth and tenth generations, respectively. Lowercase, uppercase and italic letters indicate significant differences among treatments in the same generation, and the asterisk (*) indicates significant differences between generations under complex Cd–Pb exposure (** $p < 0.01$, *** $p < 0.001$).

### 3.1.3. AHP Method Analysis of Complex Cd–Pb Exposure Affecting *S. litura*

The weighted total scores showed that complex Cd–Pb stress had positive effects on *S. litura*, and single Pb exposure showed negative effects on *S. litura*, regardless of generation (Table 1). For single Cd exposure, a positive effect, negative effect and no effect were found in *S. litura* from the first, fifth and tenth generations, respectively (Table 1).

**Table 1.** Index evaluation score by the analytic hierarchy process (AHP) method.

| Generation | Treatments | Larval Survival | Survival Pupation Rate | Eclosion Rate | Larval Weight | Pupal Weight | Adult Weight | AD | ECI | ECD | RCR | RGR | Weighted Score | Rank |
|---|---|---|---|---|---|---|---|---|---|---|---|---|---|---|
| The first generation | CK | 5 | 5 | 5 | 3 | 5 | 5 | 1 | 3 | 5 | 1 | 3 | 3.96 | 3 |
| | M | 5 | 5 | 4 | 3 | 5 | 5 | 1 | 5 | 5 | 1 | 5 | 4.13 | **1** |
| | Cd | 5 | 5 | 4 | 5 | 5 | 5 | 1 | 4 | 5 | 1 | 3 | 4.12 | 2 |
| | Pb | 5 | 5 | 3 | 3 | 5 | 5 | 1 | 4 | 5 | 1 | 3 | 3.81 | 4 |
| The fifth generation | CK | 5 | 5 | 5 | 5 | 4 | 3 | 2 | 4 | 4 | 4 | 5 | 4.30 | 2 |
| | M | 5 | 5 | 5 | 5 | 5 | 5 | 3 | 5 | 5 | 5 | 3 | 4.72 | **1** |
| | Cd | 5 | 5 | 5 | 5 | 1 | 3 | 1 | 3 | 3 | 1 | 5 | 3.58 | 4 |
| | Pb | 5 | 5 | 5 | 5 | 2 | 3 | 2 | 5 | 4 | 3 | 5 | 4.10 | 3 |
| The tenth generation | CK | 5 | 5 | 5 | 5 | 3 | 3 | 1 | 3 | 3 | 1 | 5 | 3.78 | 2 |
| | M | 5 | 5 | 5 | 5 | 5 | 5 | 1 | 5 | 5 | 3 | 1 | 4.30 | **1** |
| | Cd | 5 | 5 | 5 | 5 | 3 | 3 | 1 | 3 | 4 | 2 | 3 | 3.78 | 2 |

Treatments, CK, M, Cd and Pb denoted *S. litura* larvae were exposed to 0, 4.06 mg/kg Cd + 12.5 mg/kg Pb, 4.06 mg/kg Cd and 12.5 mg/kg Pb, respectively.

### 3.2. Cd Ingestion, Excretion, and Accumulation by S. litura

After *S. litura* were fed on complex Cd–Pb or single Cd treated diets, Cd concentrations in insect bodies and excreted matter significantly increased (Figure 3A). There were no remarkable differences of Cd concentrations in larvae, prepupae, pupae, feces, and larval and prepupal exuviates were found between the complex Cd–Pb and single Cd treatments, regardless of the generation (Figure 3A). For the adults from the first and fifth generations, the remarkably higher Cd concentrations were detected in the single Cd treatment rather than in the complex Cd–Pb treatment, and the opposite was observed in puparium.

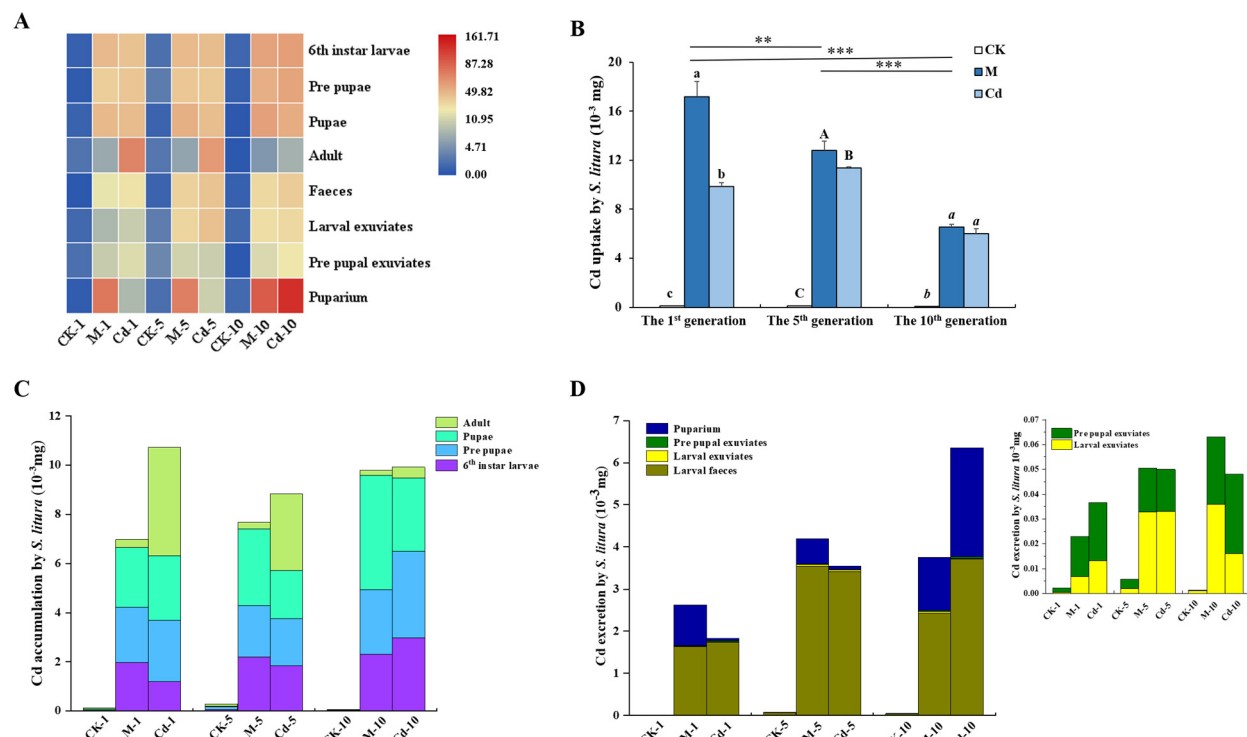

**Figure 3.** Cd uptake, accumulation and excretion by *S. litura* exposed to complex Cd–Pb and single Cd. Cd concentrations (mg/kg) in insect bodies and excreta (**A**). Cd uptake by *S. litura* (**B**). Cd accumulation by *S. litura* (**C**). Cd excretion by *S. litura* (**D**). Lowercase, uppercase and italic letters indicate significant differences among treatments in the same generation, and the asterisk (*) indicates significant differences between generations under complex Cd–Pb exposure (** $p < 0.01$, *** $p < 0.001$).

Cd uptake, accumulation and excretion by *S. litura* are shown in Figure 3B–D. For the first and fifth generations, significantly higher Cd uptake and Cd accumulation were found in larvae exposed to complex Cd–Pb stress compared to Cd stress alone (Figure 3B,C). However, significantly lower Cd accumulation from the tenth generation was found in larvae exposure to complex Cd–Pb compared to Cd alone. In addition, Cd uptake by larvae from the complex Cd–Pb treatment significantly decreased over generations (Figure 3B,C). Significantly lower Cd accumulation in prepupae from the tenth generation was found in *S. litura* larvae fed on complex Cd–Pb treated diets rather than on single Cd treated diets (Figure 3C, Supplementary Data Table S7). Significantly higher Cd accumulation in pupae from the fifth and tenth generations was found in complex Cd–Pb treatment compared to single Cd treatment. Significantly, lower Cd accumulation in adults was found in *S. litura* larvae fed on the complex Cd–Pb treated diet compared to the single Cd treated diets, regardless of the generation (Figure 3C).

Cd excretion by *S. litura* via the feces was the highest, the puparium was the second, and the lowest was the exuviates (Figure 3D, Supplementary Data Table S8). Significantly lower Cd excretion via the feces from the tenth generation was found in larvae exposed to the complex Cd–Pb than to the Cd alone, and the contrasting case was found in the

exuviates of larvae. For the puparium from the first and fifth generations, the remarkably higher Cd excretion was found in the complex Cd–Pb treatment rather than in the single Cd treatment, and the opposite was found in the puparium from the tenth generation. Generation, treatment, and their interactions were shown to have a strong significant impact on Cd concentrations in samples and Cd ingestion, excretion, and accumulation by *S. litura* (Supplementary Data Tables S6 and S11).

### 3.3. Pb Ingestion, Excretion, and Accumulation by S. litura

After *S. litura* were exposed to complex Cd–Pb or Pb alone, Pb concentrations in the insect body and excreted matter significantly increased (Figure 4A). Regardless of whether *S. litura* were from complex Cd–Pb or single Pb treatments, the highest Pb concentrations were detected in the pupae, the prepupae and larvae were the second, and the lowest was found in adults. In addition, the highest Pb concentrations were detected in puparium, the second in feces and larval exuviates, and the lowest was determined in prepupal exuviates.

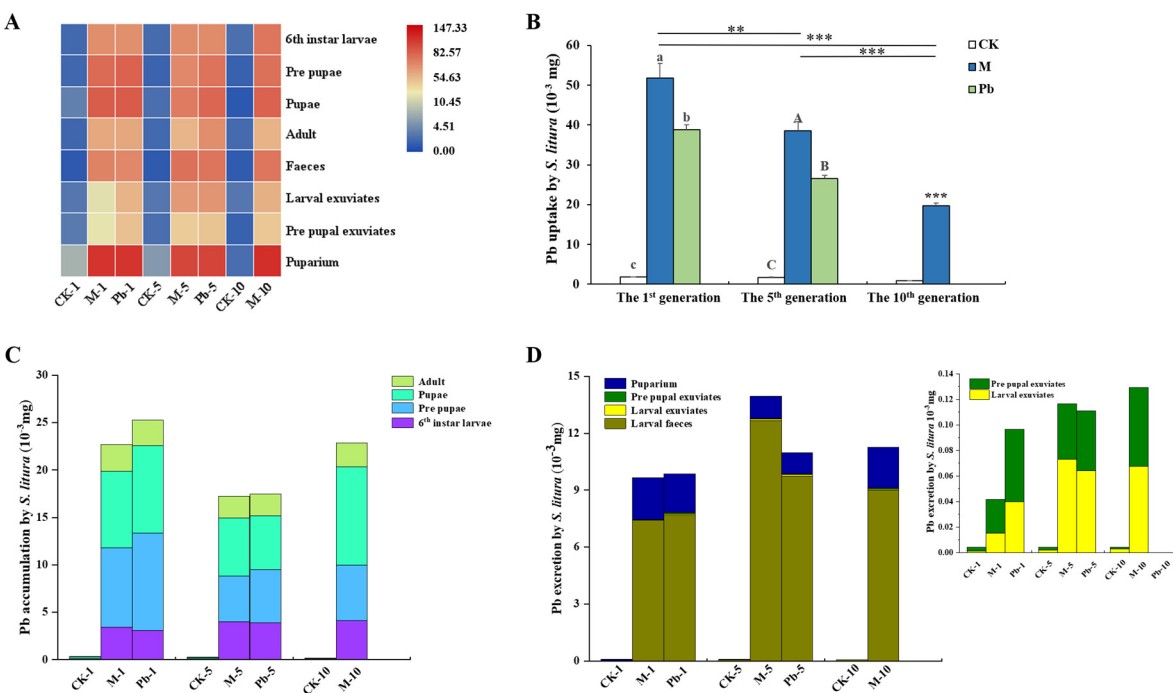

**Figure 4.** Pb uptake, accumulation and excretion by *S. litura* exposed to complex Cd–Pb and single Pb. Pb concentrations (mg/kg) in insect bodies and excreta at different developmental stages (**A**). Pb uptake by *S. litura* (**B**). Pb accumulation by *S. litura* (**C**). Pb excretion by *S. litura* (**D**). Lowercase, uppercase and italic letters indicate significant differences among treatments in the same generation, and the asterisk (*) indicates significant differences between generations under complex Cd–Pb exposure (** $p < 0.01$, *** $p < 0.001$).

Pb uptake, accumulation and excretion by *S. litura* are shown in Figure 4B–D. Pb uptake by the larvae significantly decreased over generations; significantly higher Pb uptake was found in larvae exposed to complex Cd–Pb than to single Pb (Figure 4B). Pb accumulation in larvae and adults from complex Cd–Pb treatment were similar with those from single Pb treatment, regardless of the generation (Figure 4C, Supplementary Data Table S9). Significantly lower Pb accumulation in pupae from the first generation was found in the complex Cd–Pb treatment rather than the single Pb treatment (Supplementary Data Table S9). For *S. litura* from the first generation, no significant difference in Pb excretion via larval feces, exuviates, and puparium was observed between complex Cd–Pb treatment and single Pb treatment (Figure 4D, Supplementary Data Table S10). Generation and treatment were shown to have a strong significant impact on Pb content in the samples, Pb ingestion,

excretion and accumulation by *S. litura*, except for Pb concentrations and accumulation in adults (Supplementary Data Tables S6 and S11).

### 3.4. Correlation of Fitness Parameters with Cd or Pb Accumulation

No significant correlation was found between Cd accumulation in the insect body and insect weight at the developmental stages (expect for pupae from the tenth generation), regardless of generations (Figure 5). A significantly positive correlation between Cd accumulation in larvae and ECI, ECD and RGR was found in *S. litura* from the first generation. RGR of larvae from the fifth generation was negatively correlated with Cd accumulation. For the tenth generation, a significantly positive correlation was observed between Cd accumulation and ECI or ECD; a Cd accumulation was negatively correlated with AD or RGR.

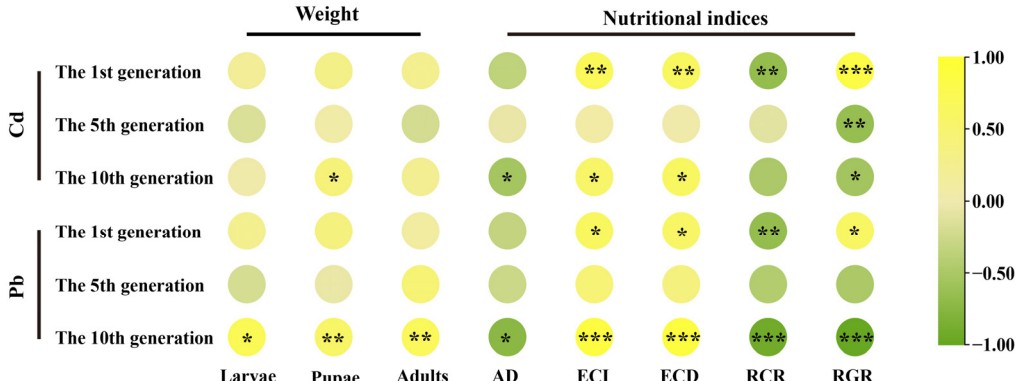

**Figure 5.** Correlation of fitness parameters with Cd or Pb accumulation. * Represents significance at the 0.05 probability level, ** represents significance at the 0.01 probability level, *** represents significance at the 0.001 probability level.

The correlations between Pb accumulation in the insect body and health parameters of the insect were also analyzed (Figure 5). For the tenth generation, Pb accumulation by insects and insect weight showed a significantly positive correlation; a significantly positive correlation was observed between Pb accumulation and ECI or ECD; Pb accumulation was negatively correlated with AD, RCR or RGR. For the first generation, Pb accumulation was positively correlated with ECI, ECD or RGR, while Pb accumulation was negatively correlated with RCR.

## 4. Discussion

Recent studies suggested that binary mixtures of heavy metals have inhibitory effects on organisms [5], demonstrating additive [49], synergistic [50] or antagonistic interactions [26,51]. Concurrent exposure to 0.5 mg/L mercury (Hg) and 0.2-2 mg/L Cd induced a delay of larval–pupal transitions in *Drosophila melanogaster*, confirming joint effects between Cd and Hg [50]. The silkworm (*Bombyx mori*) fed Pb and Cd mixtures had a larger body size than when fed Cd alone [52], showing an antagonistic response. The combination of zinc (Zn) and Cd drastically reduced the growth and survival of the aphids *Myzus persicae* [53], *Pieris napi* and *Athalia rosa* [54], and *Plutella xylostella* [55], displaying an additive interaction. However, our study demonstrated that the complex Cd (4.06 mg/kg Cd) and Pb (12.5 mg/kg Pb) promoted positive effects on *S. litura* based on the weighted scores of life history traits by AHP, regardless of the generation (Table 1). Similarly, Song et al. suggested that, compared with exposure to Cd alone, the combination of Cd and Zn at a low concentration could significantly promote the growth and development of *Sitobion avenae* [56]. In addition, antagonistic effects between Pb and Cd were found in this study when compared with insects that were exposed to Pb or Cd alone (Table 1). Di et al. also indicated that the combination of Cu and Cd demonstrated an initially antagonistic effect on *Apis mellifera* larvae at low doses but changed to a strongly synergistic effect at higher

concentrations [9]. In contrast, Fargašová concluded that the toxicity of binary combinations of Cd, Cu, Zn, Al to 3rd to 4th larvae of *Chironomus plumosus* showed synergism at the lower concentrations, and an antagonistic effect was confirmed when insects were exposed to binary combinations at higher concentrations [6].

In this study, *S. litura* exposed to Cd–Pb stress for five generations ingested more Cd via their diet (Figure 3B), and they had significantly higher ECI and ECD (representing the efficiency of converting assimilated foods into insect biomass, [57]) when compared with insects that were exposed to Cd alone. This suggested that *S. litura* gained more energy from food ingestion. However, Cd accumulation by the insects exposed to Pb-Cd mixtures was remarkably less than that of insects exposed to Pb or Cd alone (Figure 3C), which was due to significantly greater Cd excretion via the feces and puparium (Figure 3D). Moreover, the lower weight in the Pb-Cd treatment group was comparable to that of Cd alone treatment (Figure 1D). Hence, *S. litura* allocates more energy for toxicant excretion and detoxification than the absorption and transformation of nutrients to biomass. Bednarska et al. also demonstrated that the decrease in the body weight might be due to the avoidance of highly contaminated food and the increased detoxification cost [58]. This indicated that the surviving insects under the Pb-Cd treatment developed tolerance to Pb-Cd stress, whereas the detail mechanisms (e.g., changes in detoxification enzymes and expression of genes related to stress tolerance) are worthy of further study. Although insects can reduce heavy metal internal body concentrations via heavy metal excretion in the feces [58], they regulate the amounts of nonessential metals (e.g., Cd and Pb) less efficiently than nutritional metals, such as Zn [58,59]. When *S. litura* larvae were exposed to Zn below the threshold level, insect RCR significantly increased, while ECD and ECI were decreased, indicating that insects feed more to compensate for less nutritional value and supply more energy for growth [46]. Conversely, the RCR of *S. litura* significantly decreased, while the ECD and ECI increased when insects were exposed to Pb-Cd mixtures (Figure 2), suggesting that despite insects ingesting more heavy metals via their diets, they allocate more energy for heavy metal excretion. These results suggested that insects had different ways to deal with essential or nonessential metals.

When the positive effects of Pb-Cd mixtures on *S. litura* among the three tested generations were compared, a significant difference in the total weighted score of life-history traits between the Pb-Cd treatment and the control group was observed in the tenth generation (Table 1). Moreover, the intake of heavy metals decreased while the excretion of heavy metals increased with the subsequent generations (Figures 3 and 4), which suggested that the tolerance of insects to Pb-Cd mixtures at low concentrations increased with generations. After being exposed to heavy metals for several generations, aquatic microfauna like *Daphnia magna*, *Limnodrilus hoffmeisteri*, *Tisbe holothuriae*, *Chironomus plumosus* and *Culicoides furens* also had metal tolerance [60]. In this study, RCR of larvae exposed to complex Cd–Pb for five and ten generations was significantly lower than that of other treatments, whereas ECI and ECD from the tenth generation significantly increased, indicating that changes in food uptake, digestion and absorption play a crucial role in the development of heavy metal tolerance. Additionally, total antioxidant capacity of the organism was enhanced along with generations [61], since activity of detoxification enzymes (e.g., glutathione transferase GST, tacrine hydrochloride hydrate AChE) rose and expression of genes (e.g., heat shock protein *HSP70*) related to stress tolerance upregulated [37,59,61]. Some studies indicated that pollutant exposure might change the life history of the organisms, which in turn could have an adaptive importance. In fact, changes in life history require more metabolic energy and may slow down other activities, which may cause the changes or decreases in the population growth rate with the size of individuals [60]. *Anopheles gambiae* shows the potential to develop heavy metal tolerance with a significant biological cost, which can adversely affect its ecological fitness [62]. Lower and lower RGR over increasing generations were found in *S. litura* exposed to complex Cd–Pb in this study which is consistent with the above case, indicating that insects had heavy metal tolerances and can allocate more energy for detoxification than growth and reproduction. However, long-lasting (across many genera-

tions) exposure of animals to metals may result in adaptive mechanisms that enhance the ability to survive with minimal detoxification costs [36,37]. Song et al. reported that insects under low heavy metal concentrations can produce a "toxicant excitatory effect", exerting a metabolic compensatory mechanism [56]. Tolerance development may be due to the physiological changes resulting in lowered metabolic rates or may be because of the production of metallothionein-like proteins [63], but the exact mechanism by which this occurs is not understood. The molecular mechanism of this enhanced long-term tolerance also deserves further study. In addition, the short-term effects of single Cd stress (4.06 mg/kg Cd) or Pb stress (12.5 mg/kg Pb) on *S. litura* were positive or negligible, whereas the negative effects on *S. litura* were long-term (five and ten generations). Heavy metal accumulation in larvae and RGR showed a positive correlation for the first generation but a negative correlation for the fifth and tenth generations. Hence, there is a need to assess the long-term and cumulative effects of complex heavy metal exposure on insects at low concentrations.

Evaluation of the impacts of complex heavy metal exposure at low concentrations on herbivorous insects is of particular importance to the ecological and eco-toxicological risk to terrestrial food chains. In addition, some studies showed that the development of insect heavy metal tolerance probably led to the increase in their tolerance to other stresses (e.g., pesticides) [29,40,41], thus the measures of pest management in the farmland with mild or moderate heavy metal pollution should be changed. In this study, we only examined the effects of complex Cd–Pb exposure on herbivorous insects under standardized conditions that were beneficial to a testing environment but not always reflective of natural communities [64]. Indeed, heavy metal pollution in the field is the coexistence of multiple heavy metals, and their impacts on organisms were more diversified and complicated [4,5,29]. It is more meaningful to explore the tolerance mechanism of insect populations to heavy metal exposure using indoor work and field investigations simultaneously, which will be conducted in our future research.

## 5. Conclusions

The complex Cd–Pb exposure (4.06 mg/kg Cd + 12.5 mg/kg Pb) under continuous multigenerations had positive effects on *S. litura*, reflected in the higher weighted scores calculated by the AHP method. Compared with Cd (4.06 mg/kg Cd) and Pb (12.5 mg/kg Pb) exposure alone, complex Cd–Pb exposure showed antagonistic effects. After exposure to complex Cd–Pb, although larvae significantly increased heavy metal uptake along with an elevated ECD and ECI, *S. litura* enhanced heavy metal excretion via the feces and puparium; consequently, heavy metal accumulation in the insect body significant decreased. With increasing generations, the progressive effects of complex Cd–Pb exposure on *S. litura* became increasingly obvious, and the intake of heavy metals decreased, while the excretion of heavy metals increased. These results suggested that the tolerance of insects to Cd–Pb mixtures at low concentrations increased with generations. However, exact mechanisms of insect tolerance development to heavy metals after long-term exposure are not yet clarified and are worthy of further study.

**Supplementary Materials:** The following supporting information can be downloaded at: https://www.mdpi.com/article/10.3390/agronomy13051207/s1, Text S1: The rearing conditions of *S. litura*; Text S2: *S. litura* bioassay with complex Cd–Pb exposure for the first, fifth and tenth generations; Text S3: Determination of heavy metal concentrations in different samples; Table S1: The ingredients of artificial diets; Table S2: Formulae for calculating the life parameters and nutritional indices of *S. litura*; Table S3: Index evaluation system; Table S4: The scoring criteria of indicators; Table S5: Interaction of treatment and generation on life history traits and food utilization of *S. litura*; Table S6: Interaction of heavy metal treatment and generation on Pb or Cd concentrations in different samples; Table S7: Cd accumulation in *S. litura* at different development stages from the first, fifth and tenth generations; Table S8: Cd excretion in *S. litura* at different development stages from the first, fifth and tenth generations; Table S9: Pb accumulation in *S. litura* at different development stages from the first, fifth and tenth generations; Table S10: Pb excretion in *S. litura* at different development stages from the first, fifth and tenth generations; Table S11: Interaction of treatment and generation on Cd or Pb ingestion,



accumulation, and excretion by *S. litura*; Figure S1: Deformity of *S. litura* pupae from the eighth generation under continuous Pb stress.

**Author Contributions:** Conceptualization, Y.S., M.Z. and J.C.; methodology, M.Z., J.C. and S.C.; software, K.W. and M.Z.; validation, M.Z., J.C. and S.C.; investigation, M.Z., J.C., K.W. and S.C.; resources, Y.S. and J.W.; writing—original draft preparation, Y.S., M.Z., K.W. and J.C.; writing—review and editing, Y.S., M.Z., J.C. and J.W.; funding acquisition, Y.S. and J.W.; All authors have read and agreed to the published version of the manuscript.

**Funding:** This research was funded by the National Natural Science Foundation of China: 32071619, the Natural Science Foundation of Guangdong Province, China: 2019A1515011998 and 2023A1515011052.

**Data Availability Statement:** The datasets used and analyzed during the current study available from the corresponding author upon request.

**Conflicts of Interest:** The authors declare no conflict of interest.

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
