# Peer review of "Complex Cd–Pb Multigenerational Exposure Improves the Growth and Food Utilization of the Cutworm Spodoptera litura"

_agronomy, doi:10.3390/agronomy13051207_

Round 1

Reviewer 1 Report

The present study evaluated to what extent complex Cd-Pb stress had effects on the growth of Spodoptera litura in comparison with the stress of Cd alone or Pb alone. Surprisingly, the authors found that feeding complex Cd-Pb to the larvae had positive effects on S. litura, resulting in increasing the weight of pupae and adults, compared with the control, feeding Cd alone and feeding Pb alone. Feeding complex Cd-Pb resultantly enhanced heavy metal excretion via the faeces and puparium and decreased the accumulation of the heavy metals in the adults. Interestingly, this study indicated that with successive generations, S. litura exhibited high tolerance to complex Cd-Pb by decreasing the uptake and increasing the excretion of complex Cd-Pb. 

I highly appreciate this study because it examined the effects of feeding of complex Cd-Pb over 10 generations and it found that the tolerance of S. litura to complex Cd-Pb was enhanced with successive generations. However, I am not satisfied that the authors did not discuss the reason why tolerance to complex Cd-Pb was enhanced with generations. Discussion about this factor is absolutely needed. 

In general, it can be supposed that due to natural selection, S. litura individuals that have higher ability to excrete the heavy metals or ability not to uptake them have been selected from generation to generation. I want to know whether the authors agree that natural selection is the main factor for the increased tolerance to complex Cd-Pb or assume other mechanisms. In relation to this problem, I want to know the origin of S. litura individuals used for the experiments; were they collected from the wild or maintained in the laboratory for a long time? In addition, how the authors chose males and females for copulation to make the next generation. These problems were related to the action of natural selection in the laboratory, so clarify these points.

     The authors use five nutritional indices, RCR, AD, ECI, ECD, and RGR (L86-L88). However, there are no detailed information about these indices. Clarify how the authors calculated these indices. I want to give a critical comment on the use of these indices when the authors use them in comparison among treatments and generations. It is because if these indices include weight as a component, they have any correlation with the weight of larvae, pupae or adults. 

Minor comments are as follows,

Line 68. “are described in [29]” may be better.

Line 84. “nutritional” may be “Nutritional”.

Line 158-169. Abbreviations (AD, ECD and so on) were difficult to understand. Do not use abbraviations.

Figure 2 legenda. Do not use abbreviations (AD, ECI, ECD, RCD, and RGR). They are difficult to understand.

Line 199. “lower” may be wrong. I think that the opposite is true in Fig. 3C, so “Significantly higher Cd accumulation” may be correct.

Line 206. I cannot see this result from Fig. 3D, in which the values of larval exuviates are invisible.

Line 228-230. I cannot see this result from Fig. 4C. Is there actually statistical significance?

Line 231. “via larval faeces and exuviates, puparium” should be “via larval faeces, exuviates, and puparium”.

Line 272. The authors said “promoted effects”. I want to know “promoted positive effects” or “promoted negative effects”. Clarify this point.

 ---

Reviewer 2 Report

The work by Meifang Zhang and colleagues show in this study, the single Cd stress, single Pb stress and complex Cd-Pb stress to feed S. litura for 10 continuous generations under laboratory conditions. to investigate Cd or Pb ingestion, excretion and accumulation in S. litura at the different developmental stages and analyze their combined effects on growth and food utilization after exposure to complex Cd-Pb stress in the long term.

I have found some minor details in the ms that need to be revised, which hopefully will improve the quality of the present work.

1)              Keywords: Please do not include words mentioned in the title of the ms, and Sort key-words alphabetically. 2)             In the description of the figures, abbreviations should not be used. 3)           Line 77 : give more details on the number of larvae per dish, the diameter of the dishes used and the incubation conditions.

4)               Line 88 : Why 14 larvae were treated and reared for each treatment.

5)               Line 80 : how did you measure the adults body.

6)               Why you didn't analyse the detoxification enzymes of diferents stage of insect

Reviewer 3 Report

L39: In key contribution, explaining what positive effects refers to would be better.

In the introduction section, what are the possible sources of exposure to heavy metals? Arguments should be built on why this is important to assess heavy metals contaminations in insects.

L40: Replace "Problem" by "Concern"

L56: Add authority name of Spodoptera litura

L68: Rephrase the statement. Include the source and add a brief mention of rearing...

L69: Please mention the ingredients of the artificial diet used and the larvae fed how often, and how adults were fed. What were the experimental conditions under which rearing was done?

L89: Rephrase this statement.

In the statistical analyses portions, I would suggest to arc-sine square transform the percentage data before analysis.

L127: 3.1.1. Impacts of complex Cd-Pb stress on the growth of S. litura.  Line 128 discusses survival rates whereas the heading reads "the growth" Please be consistent and revise the results in lines 129-131 for clarity and understandability. What is meant by eclosion?

Implications of current results in the typical context of pest management should be made a part of the discussion. What are the limitations of the current study...

Overall this is a good study. I would suggest authors should add more on the significance of this study in the typical context of pest control or tri-trophic relationships. Some suggestions in the statistical analyses part are recommended. Moreover, the Insect rearing part and the bioassay procedure are recommended to revise. 

Round 2

Reviewer 3 Report

The manuscript has been revised appropriately.